# EVENT-ANCHORED FRAME SELECTION FOR EFFICIENT LONG-VIDEO UNDERSTANDING

## ABSTRACT

Massive frame redundancy and limited context window make efficient frame selection crucial for long-video understanding with large vision-language models (LVLMs). Prevailing approaches, however, adopt a flat sampling paradigm which treats the video as an unstructured collection of frames. In this paper, we introduce **E**vent-Anchored **F**rame **S**election (**EFS**), a hierarchical, event-aware pipeline. Leveraging self-supervised DINO embeddings, EFS first partitions the video stream into visually homogeneous temporal segments, which serve as proxies for semantic events. Within each event, it then selects the most query-relevant frame as an anchor. These anchors act as structural priors that guide a global refinement stage using an adaptive Maximal Marginal Relevance (MMR) scheme. This pipeline ensures the final keyframe set jointly optimizes for event coverage, query relevance, and visual diversity. As a **training-free, plug-and-play module,** EFS can be seamlessly integrated into off-the-shelf LVLMs, yielding substantial gains on challenging video understanding benchmarks. Specifically, when applied to LLaVA-Video-7B, EFS improves accuracy by **4.7%, 4.9%, and 8.8%** on VideoMME, LongVideoBench, and MLVU, respectively. Code is provided in the supplementary material and will be released publicly.

## 1 INTRODUCTION

The remarkable success of transformer-based large language models (LLMs) in natural language processing (Achiam et al., 2023) has catalyzed their extension into multimodal domains. By integrating powerful visual encoders with advanced reasoning capabilities of LLMs, Large Vision-Language Models (LVLMs) (Liu et al., 2023; Wang et al., 2024) have emerged, rapidly advancing machine perception. Leveraging pre-training on large-scale video-text datasets, LVLMs are now capable of complex video tasks such as detailed video captioning (Chen et al., 2024), nuanced question answering (Min et al., 2024), and sophisticated temporal reasoning (Qian et al., 2024).

Despite these successes, LVLMs confront a fundamental bottleneck with long-form videos: the massive number of frames clashes with limited computational budget and fixed context windows. Directly processing every frame is simply infeasible. This necessitates a mechanism to condense the vast visual stream into a concise yet representative input. Consequently, frame selection has become a critical, pragmatic prerequisite for applying LVLMs to real-world long-form content. While alternative strategies like extending context windows (Liu et al., 2024; Zhang et al., 2024b) or video-to-text summarization (Ma et al., 2025) exist, they often incur prohibitive computational overhead or significant information loss, making efficient frame selection a more direct and balanced approach.

However, the effectiveness of frame selection depends entirely on how frames are selected. Many existing techniques (Park et al., 2024; Yu et al., 2024; Sun et al., 2025; Huang et al., 2025; Fang et al., 2025; Zhang et al., 2025; Xu et al., 2025) utilize a flat sampling paradigm, treating the video as an unstructured collection of frames and disregarding higher-level semantic relationships. This paradigm is fundamentally temporally-agnostic, overlooking the intrinsic narrative and event structure of video. Consequently, these methods often fail to select a frame set that simultaneously and effectively balances the three essential pillars of ideal selection: query relevance, comprehensive event coverage, and sufficient visual diversity. As illustrated in Figure 1(a), a flat sampling strategy is prone to missing crucial events, leading LVLMs to reach incorrect conclusions.

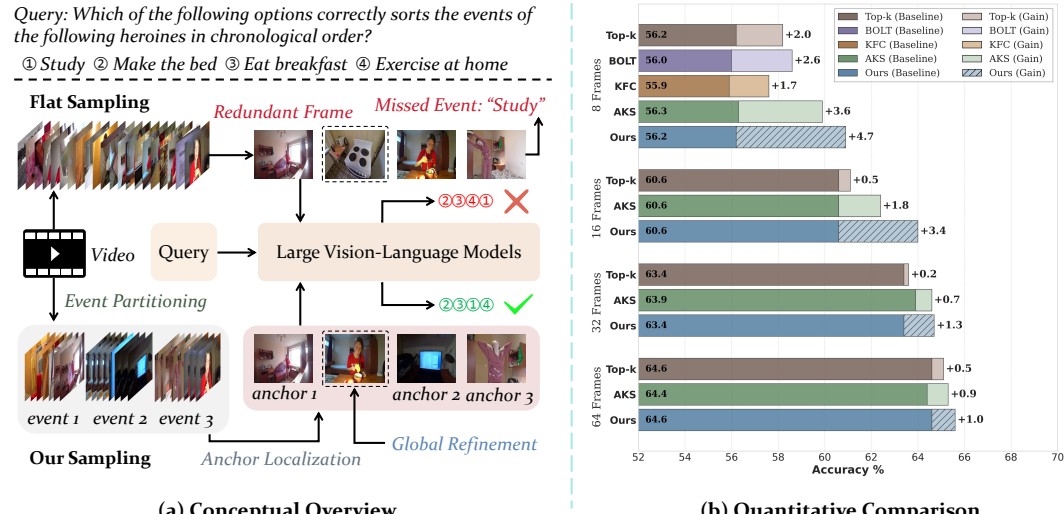

**(a) Conceptual Overview**                    **(b) Quantitative Comparison**

Figure 1: **Illustration of how our Event-Anchored Frame Selection (EFS) outperforms flat sampling.** (a) A conceptual overview showing that flat sampling yields redundant frames and misses key events, causing LVLMs to fail, while EFS partitions videos into events, selects query-relevant anchors, and performs anchor-guided global refinement to enable correct chronological reasoning. (b) A quantitative comparison on VideoMME using LLaVA-Video-7B, where EFS consistently delivers the largest accuracy gains across 8/16/32/64-frame budgets over different sampling strategies. The text to the left of each bar represents the baseline performance for that method, while the text on the right indicates the corresponding gain. Further details are provided in Section 4.2.

To overcome this limitation, we propose shifting from a temporally-agnostic to an event-aware perspective. We introduce Event-Anchored Frame Selection (EFS), a hierarchical, training-free pipeline grounded in the video's event structure. Our core insight is that an optimal keyframe set must be built upon a macroscopic understanding of the video's narrative flow. EFS first partitions the video into visually coherent temporal segments, serving as reliable proxies for semantic events. It then establishes a structural "backbone" by selecting the most query-relevant frame from each event as an anchor. This set of event anchors then guides a global refinement stage, which leverages an adaptive Maximal Marginal Relevance (MMR) scheme to enrich the selection with diverse, informative frames. This hierarchical process ensures the final keyframe set jointly optimizes for event coverage, query relevance, and visual diversity.

As a plug-and-play module, EFS can be seamlessly integrated with off-the-shelf LVLMs without any additional training. We validate EFS on three widely used long-video question-answering benchmarks: VideoMME (Fu et al., 2025), LongVideoBench (Wu et al., 2024), and MLVU (Zhou et al., 2024). Across all benchmarks, EFS delivers consistent and substantial performance gains. Specifically, when applied to LLaVA-Video-7B and LLaVA-OneVision-7B, EFS improves accuracy by 4.7%, 4.9%, and 8.8%, and by 3.3%, 6.2%, and 8.8% on VideoMME, LongVideoBench, and MLVU, respectively. These results demonstrate that event-aware frame selection is essential for unlocking the full potential of LVLMs in long-video understanding.

Our main contributions can be summarized as follows:

- We propose Event-Anchored Frame Selection (EFS), a novel, training-free hierarchical framework that perceives a video's macroscopic event structure before performing fine-grained frame selection, surpassing the limitations of flat, temporally-agnostic sampling.

- We design an anchor-guided global refinement strategy that employs an adaptive MMR scheme. This method dynamically calibrates the diversity threshold based on the video's own content statistics, enhancing robustness and adaptability across different video types.

- We conduct extensive experiments on three challenging long-video understanding benchmarks and demonstrate that EFS significantly boosts the reasoning performance of existing LVLMs, clearly underscoring the value of event-aware frame selection.

## 2 RELATED WORK

**LVLMs for Long Video Understanding.** Despite impressive performance on short clips (Maaz et al., 2024; Lin et al., 2024), LVLMs face significant challenges with long-form videos due to massive frame redundancy and limited context windows. To mitigate this, one line of research focuses on architectural modifications, such as extending the context window for the model (Liu et al., 2024; Zhang et al., 2024b) or reducing visual tokens by pruning and merging (Shen et al., 2024; Luo et al., 2025; Li et al., 2024b). However, these approaches often suffer from high computational overhead or risk discarding critical visual details. Another strategy involves converting video content into text to leverage the language capabilities of LLMs (Park et al., 2024; Ma et al., 2025; Fan et al., 2024; Luo et al., 2024). This method, while efficient, still leads to significant information loss, constraining the understanding of visual nuances.

**Frame Selection for Video Understanding.** Frame selection provides a practical alternative to the approaches above. Traditional methods like uniform sampling or early keyframe extraction (Nasreen & Dr Shobha, 2013; Sheena & Narayanan, 2015) are often query-agnostic and insufficient for complex reasoning tasks. Recent work explores query-based frame selection, ranking frames by relevance to specific queries. For example, BOLT(Liu et al., 2025a) applies inverse transform sampling, while Frame-Voyager(Yu et al., 2024) uses a ranking-based pipeline for optimal frame combinations. Other notable methods include mDP$^3$(Sun et al., 2025), which formulates frame selection as a Markov decision process combined with determinantal point processes, AKS (Tang et al., 2025), which jointly optimizes for both query relevance and video coverage, and Nar-KFC(Fang et al., 2025), which models the problem as subgraph selection. Although these approaches can improve query relevance or diversity, they predominantly adopt a flat sampling paradigm, treating video as an unstructured frame sequence and ignoring event structure, which limits event coverage. In contrast, our hierarchical event-aware framework jointly optimizes relevance, coverage, and diversity.

## 3 METHODOLOGY

In this section, we elaborate on our Event-Anchored Frame Selection (EFS) framework. As illustrated in Figure 2, EFS first obtains a macroscopic understanding by partitioning the video into coherent visual events. This events structure then guides a global refinement process to produce a keyframe set optimized for relevance, coverage, and diversity.

### 3.1 PROBLEM FORMULATION

Given a long video of $T$ frames, $V = \{I_1, I_2, ..., I_T\}$, and a user query $Q$, the task of keyframe selection is to extract a concise yet representative subset of frames $K = \{I_{t_1}, I_{t_2}, ..., I_{t_k}\}$, where the indices are chronologically ordered and $k << T$. The value of $k$ is typically pre-defined based on both the context window limitations of LVLMs and the requirements of downstream tasks or user preferences. The objective is to select $K$ so that, when it is provided as visual context along with the query $Q$, an LVLM can generate the most accurate and comprehensive answer possible. This requires balancing three core objectives: ensuring thorough coverage of key events, maintaining strong relevance to the query, and promoting sufficient diversity to minimize redundancy.

However, the most strategy in existing LVLM-based long-video understanding pipelines is uniform sampling. While simple and efficient, it largely disregards the intrinsic temporal and semantic structure of the video, often missing critical, query-relevant events and introducing redundant or uninformative frames. Hence, more intelligent, structure-aware keyframe selection frameworks are needed.

### 3.2 VISUAL & SEMANTIC SIGNAL ACQUISITION

We begin by sampling the video at 1 frame-per-second, yielding a length-$N$ candidate frame sequence $\mathcal{I} = \{I_1, I_2, ..., I_N\}$. In this sequence, we extract two fundamental signals for each frame.

**Image-Text Matching.** To quantify the semantic relevance of each frame to the query, we use the Image-Text Matching (ITM) head of Blip2 (Li et al., 2023) (referred to as BLIP2-ITM), which is designed to output a score indicating the alignment between a given image and a text description.

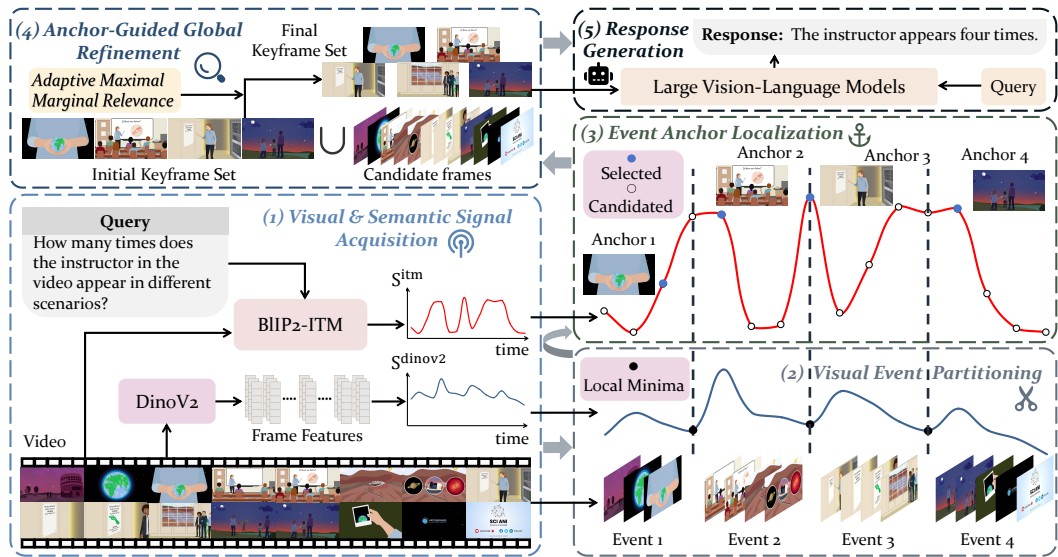

Figure 2: An illustration of our Event-Anchored Frame Selection (EFS) pipeline. Given a user query, such as "How many times does the instructor appear in different scenarios?", EFS operates in distinct stages: (1) It first acquires visual signals (from DinoV2) and semantic relevance scores (from BLIP2-ITM). (2) The visual signals are used to partition the video into coherent events at local minima of temporal similarity. (3) Within each event, it localizes the most query-relevant frame as an anchor. (4) This initial set of anchors is then globally refined via Adaptive MMR to form the final keyframe set. (5) Finally, this curated set enables the Large Vision-Language Models to accurately answer the query, correctly identifying the four appearances.

This score serves as an excellent measure of query relevance. For each frame $I_i$ and the text query $Q$, the relevance score is computed as: $s_i^{\text{itm}} = \text{BLIP2-ITM}(I_i, Q)$, These scores, forming a relevance signal $\{s_1^{\text{itm}}, \ldots, s_N^{\text{itm}}\}$, allow us to prioritize frames that pertain to the user's information need.

**Temporal Similarity Calculation.** To understanding the temporal structure of the video, we measure visual similarity between frames using DINOv2 (Oquab et al., 2023), a powerful self-supervised model known for its robust visual embeddings that capture both high-level semantic content and low-level structural details without requiring explicit labels. Its features are particularly effective at discerning meaningful changes in visual content. For each frame $I_i$, we extract its DINOv2 feature vector, $\mathbf{f}_i \in \mathbb{R}^d$, and L2-normalize it. Using these embeddings, we then compute an inter-frame temporal similarity score, $s_i^{\text{dinov2}}$, for each frame $i$ by comparing it to its neighbors within a weighted sliding window of size $l$. This captures the local visual coherence:

$$s_i^{\text{dinov2}} = \frac{\sum_{j \in \mathcal{N}(i)} w_{|i-j|} \cdot \cos(\mathbf{f}_i, \mathbf{f}_j)}{\sum_{j \in \mathcal{N}(i)} w_{|i-j|}},$$

where $\mathcal{N}(i)$ is the neighborhood of frame $i$, and the weight $w_d$ decreases linearly as the distance $d = |i-j|$ increases. Based on empirical design, we set $l$ to 3; detailed ablation studies can be found in the Appendix D.2. The resulting inter-frame temporal similarity curve, $\{s_1^{\text{dinov2}}, \ldots, s_N^{\text{dinov2}}\}$, reveals the rhythm of visual changes throughout the video.

### 3.3 EVENT-ANCHORED FRAME SELECTION

Our EFS method follows a three-stage, coarse-to-fine process: partitioning the video into events, localizing an anchor in each event, and performing a globally-aware refinement.

**Visual Event Partitioning.** We conceptualize "events" as visually homogeneous temporal segments. In video production, a significant change in visual content—such as a camera cut or a major shift in scenery—typically marks the boundary between distinct scenes or events. These moments directly correspond to local minima in our temporal similarity curve $\{s_1^{\text{dinov2}}, \ldots, s_N^{\text{dinov2}}\}$, as they

signify maximum visual change. Directly selecting these minima as our initial event boundaries is highly efficient and incurs no extra overhead, unlike approaches that rely on clustering or other computationally intensive segmentation methods.

However, long videos often contain hundreds of events, exceeding the practical token budget of LVLMs. To address this, we set a target event count $M$. If the initial partitioning yields more than $M$ events, we iteratively merge the most similar adjacent pair, using the cosine similarity between their mean DINOv2 features. This continues until exactly $M$ visually distinct, temporally coherent events $\{\mathcal{G}_1, \mathcal{G}_2, \ldots, \mathcal{G}_M\}$ remain, forming the macroscopic structural for our selection.

**Event Anchor Localization & Initialization.** To ensure both complete event coverage and strong query alignment, we initialize our keyframe set by selecting one representative frame from each event. Given the importance of the user query in video understanding tasks, we select the frame with the highest query-relevance score $s^{\text{itm}}$ within each event $\mathcal{G}_j$ to serve as its "anchor". This can be formulated as

$$k_j^{\text{anchor}} = \arg\max_{i \in \mathcal{G}_j} s_i^{\text{itm}},$$

This set of anchors, $\mathcal{K}_{\text{init}} = \{k_1^{\text{anchor}}, \ldots, k_M^{\text{anchor}}\}$, forms an initial keyframe set that is firmly grounded in the video's event structure while being centered on the query's focus.

---

**Algorithm 1** Anchor-Guided Global Refinement

1: **Input:** Candidate frames $\mathcal{C}$, Anchor set $\mathcal{K}_{\text{init}}$, DINOv2 frame features $\mathbf{f}$, Relevance scores $s^{\text{itm}}$, Target size $k$, Threshold relaxation factor $\alpha$, Increment $\delta$.
2: **Output:** Final keyframe set $\mathcal{K}$.
3: $\mathcal{K} \leftarrow \mathcal{K}_{\text{init}}$; Sort $\mathcal{C}$ descending by relevance score $s^{\text{itm}}$.
4: $\mu, \sigma \leftarrow \text{mean}, \text{std}(\{\max_{I_j \in \mathcal{K}_{\text{init}}} \cos(\mathbf{f}_i, \mathbf{f}_j) \mid I_i \in \mathcal{C}\})$    // Estimate similarity distribution of $\mathcal{C}$ to $\mathcal{K}_{\text{init}}$.
5: $\theta_{\text{strict}}, \theta_{\text{loose}} \leftarrow \text{clip}(\mu - \alpha\sigma, 0, 1), \text{clip}(\mu + \alpha\sigma, 0, 1)$       // Set adaptive strict and loose thresholds.
6: $\theta_{\text{current}} \leftarrow \theta_{\text{strict}}$
7: **while** $|\mathcal{K}| < k$ **and** $\theta_{\text{current}} \leq \theta_{\text{loose}}$ **do**
8:     **for** $I_c$ in $\mathcal{C} \setminus \mathcal{K}$ **do**
9:         **if** $|\mathcal{K}| \geq k$ **then break**
10:        **end if**
11:        **if** $\max_{I_j \in \mathcal{K}} \cos(\mathbf{f}_c, \mathbf{f}_j) < \theta_{\text{current}}$ **then**
12:            $\mathcal{K} \leftarrow \mathcal{K} \cup \{I_c\}$                        // Add frame if below current threshold.
13:        **end if**
14:    **end for**
15:    $\theta_{\text{current}} \leftarrow \min(\theta_{\text{current}} + \delta, \theta_{\text{loose}})$                    // Relax threshold for next round.
16: **end while**
17: **return** $\mathcal{K}$

---

**Anchor-Guided Global Refinement.** The anchor set provides a strong but sparse representation. The final step is to refine this set by adding more frames to enhance detail and diversity. A classic approach for this is Maximal Marginal Relevance (MMR) (Carbonell & Goldstein, 1998), which aims to select items that are both relevant to a query and dissimilar to already selected items:

$$\arg\max_{I_i \in \mathcal{C}} \left[ \lambda \cdot \text{sim}(I_i, Q) - (1 - \lambda) \max_{I_j \in \mathcal{K}} \text{sim}(I_i, I_j) \right].$$

Here, $\mathcal{K}$ denotes the set of selected frames, $\mathcal{C}$ refers to the remaining frames, and hyperparameter $\lambda$ balances relevance and diversity. This method rescoring all candidates at each step, resulting in a computational complexity of about $\mathcal{O}(|\mathcal{C}||\mathcal{K}|^2)$, which can be demanding. To improve efficiency, Cheng et al. (2025) propose an approximate greedy algorithm that sort candidates by relevance and applying a fixed diversity threshold. However, this approach is not robust: a single threshold cannot adapt to varying visual pacing-what works for an action film may fail for a slow documentary.

Our central motivation is to make the diversity threshold adaptive and data-driven, guided by the intrinsic visual structure of each video. By leveraging previously generated event anchors as a statistical prior, our anchor-guided refinement strategy tailors the selection criteria to the content redundancy of the source video. This approach dynamically adjusts the diversity threshold so that visually dense segments are subjected to stricter deduplication, while sparser regions are treated

Table 1: Video-based question answering accuracy (%) of previous LVLMs on Video-MME (Fu et al., 2025), LongVideoBench (Wu et al., 2024) and MLVU (Zhou et al., 2024). Our EFS is applied to three off-the-shelf LVLMs. Frames and LLM indicate the number of video frames input to the LVLMs and the number of parameters in the LLM part, respectively.

| Model | Params | Frames | VideoMME (w.o. sub.) | | | | LongVideo Bench (val) | MLVU (m-avg) |
| | | | Short | Medium | Long | Overall | | |
| --- | --- | --- | --- | --- | --- | --- | --- | --- |
| *Proprietary LVLMs* | | | | | | | | |
| GPT-4o mini (OpenAI, 2024a) | - | 250 | 72.5 | 63.1 | 58.6 | 64.8 | - | - |
| GPT-4o (OpenAI, 2024b) | - | 384/256/0.5fps | 80.0 | 70.3 | 65.3 | 71.9 | 66.7 | 64.6 |
| *Open-Source LVLMs* | | | | | | | | |
| Video-LLaVA (Lin et al., 2024) | 7B | 8 | 45.3 | 38.0 | 36.2 | 39.9 | 39.1 | 47.3 |
| LongVA (Zhang et al., 2024b) | 7B | 128/256 | 61.1 | 50.4 | 46.2 | 52.6 | - | 56.3 |
| Video-XL (Shu et al., 2025) | 7B | 128/256 | 64.0 | 53.2 | 49.2 | 55.5 | 50.7 | 64.9 |
| Frame-Voyager (Yu et al., 2024) | 7B | 8 | 67.3 | 56.3 | 48.9 | 57.5 | - | 65.6 |
| LongVU (Shen et al., 2024) | 7B | 1fps | - | - | 59.5 | 60.6 | - | 65.4 |
| NVILA (Liu et al., 2025b) | 8B | 256 | 75.7 | 62.2 | 54.9 | 64.2 | 57.7 | 70.1 |
| LLaVA-OneVision (Li et al., 2024a) | 7B | 8 | 65.2 | 52.0 | 45.0 | 54.1 | 54.1 | 58.6 |
| **LLaVA-OneVision + EFS** | **7B** | **8** | **70.4** (+5.2) | **55.7** (+3.7) | **46.0** (+1.0) | **57.4** (+3.3) | **60.3** (+6.2) | **67.4** (+8.8) |
| Qwen2.5-VL (Bai et al., 2025) | 7B | 16 | 68.0 | 57.0 | 47.8 | 57.6 | 56.9 | 56.5 |
| **Qwen2.5-VL + EFS** | **7B** | **16** | **69.9** (+1.9) | **57.6** (+0.6) | **49.9** (+2.1) | **59.1** (+1.5) | **60.5** (+3.6) | **66.0** (+9.5) |
| LLaVA-Video (Zhang et al., 2024c) | 7B | 64 | 76.4 | 63.1 | 54.2 | 64.6 | 58.8 | 68.1 |
| **LLaVA-Video + EFS** | **7B** | **64** | **77.3** (+0.9) | **64.3** (+1.2) | **55.2** (+1.0) | **65.6** (+1.0) | **62.1** (+3.3) | **70.9** (+2.8) |

more inclusively. Such content-aware adaptation enables robust and equitable keyframe selection across diverse video types. For implementation details, please refer to Algorithm 1. Since the outer while loop typically executes only a few times in practice, the overall complexity is approximately $\mathcal{O}(\max(|\mathcal{K}||\mathcal{C}|, |\mathcal{C}|\log|\mathcal{C}|))$, making the approach suitable for practical use. With the final keyframe set $\mathcal{K}$ determined by our EFS , it is concatenated with the query $Q$ and provided as input to a pre-trained LVLM. The model subsequently generates a comprehensive response: $\mathcal{R} = \text{LVLM}(\mathcal{K}, Q)$.

## 4 EXPERIMENTS

### 4.1 EXPERIMENTAL SETTINGS

**Evaluation Benchmarks and Models.** We evaluate our method on three long-video QA benchmarks:1) VideoMME (Fu et al., 2025), consisting of 2,700 human-annotated QA pairs, with an average video duration of 17 minutes; 2) LongVideoBench (Wu et al., 2024), using the validation set containing 1,337 QA pairs, with an average video length of 12 minutes; 3) MLVU (Zhou et al., 2024), a multi-task benchmark where we focus on the multiple-choice (M-avg) task, comprising 2,174 questions spanning 7 categories and an average duration of 11 minutes per video. To fairly assess the contribution of keyframe selection, we do not leverage video subtitles in any experiments. We adopt three Large Vision-Language Models (LVLMs) as baselines: LLaVA-OneVision(Li et al., 2024a), Qwen2.5-VL(Bai et al., 2025), and LLaVA-Video (Zhang et al., 2024c), all in their 7B variants. For fair comparison, we replicate all model results under a unified experimental setup.

**Implementation Details.** To reduce computational costs, candidate frames are uniformly sampled from each video at 1 frame per second. Query-to-frame matching scores are computed using the BLIP2-ITM-ViT-g model (Li et al., 2023), while DINOv2 (Oquab et al., 2023) is used for visual feature extraction. For all tasks, we report accuracy measured by the LMMs-Eval toolkit (Zhang et al., 2024a). All experiments were conducted on an NVIDIA A800 GPU with 80 GB of memory.

### 4.2 COMPARISONS WITH STATE-OF-THE-ARTS.

**Comparisons with different LVLMs.** We first compare our method's video question answering accuracy with leading LVLMs, including proprietary models such as GPT-4o and open-source models like NVILA. These models differ in parameter scale, input frame capacity, and architecture, providing a broad evaluation of our method's generalizability. As shown in Table 1, our Event-Anchored Frame Selection (EFS) consistently and substantially improves accuracy across all models

and benchmarks. For example, when applied to LLaVA-OneVision (8 input frames), EFS boosts accuracy by 3.3% on VideoMME, 6.2% on LongVideoBench, and 8.8% on MLVU. For Qwen2.5-VL (16 frames), gains are 1.5%, 3.6%, and 9.5%, respectively. Even with the strongest baseline, LLaVA-Video (64 frames), EFS achieves improvements of 1.0%, 3.3%, and 2.8%. These enhancements allow smaller 7B-parameter models to outperform competing methods with similar computational budgets, and in some cases, even rival much larger models. For instance, LLaVA-Video-7B with EFS attains 65.6% accuracy on VideoMME, surpassing GPT-4o-mini by 0.8%. Similarly, LLaVA-OneVision with only 8 frames outperforms GPT-4o by 2.8% on MLVU with EFS, a result not possible with uniform frame selection. These findings underscore both the robust generalizability and the simple, plug-and-play benefits of EFS across a variety of models and input configurations.

**Comparisons with different frame sampling strategies.** We further conduct a head-to-head comparison with recent query-based frame sampling methods, including Top-K, BOLT (Liu et al., 2025a), KFC (Fang et al., 2025), and AKS (Tang et al., 2025). All approaches are evaluated on VideoMME using LLaVA-Video-7B as the backbone, with improvements reported over a uniform sampling baseline for different numbers of input frames. As shown in Figure 1(b), EFS delivers the largest accuracy gains across all frame counts, outperforming baselines by 4.7%, 3.4%, 1.3%, and 1.0% at 8, 16, 32, and 64 frames, respectively. The Top-K approach provides minimal gains due to redundant selection. Although strategies like BOLT, KFC, and AKS seek to increase query relevance and frame diversity, their flat sampling paradigm often neglects the video's intrinsic event structure. In summary, EFS consistently achieves the most robust improvements, underscoring its effectiveness in informative frame selection and its advantage for accurate long-video understanding.

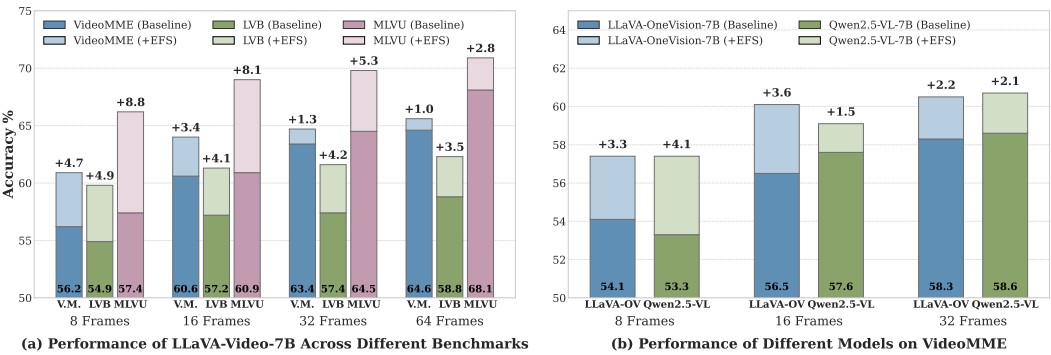

(a) Performance of LLaVA-Video-7B Across Different Benchmarks  (b) Performance of Different Models on VideoMME

Figure 3: Video-based question answering accuracy (%) with different numbers of input frames.

### 4.3 ABLATION AND ANALYSIS

**Various number of keyframes.** To assess the generalizability of EFS, we evaluate its performance across varying frame budgets and multiple LVLMs, as illustrated in the Figure 3. On LLaVA-Video-7B (subplot a), EFS consistently outperforms uniform sampling on all three benchmarks, with especially large gains under tight budgets: using only 8 frames, accuracy improves by +4.7% on VideoMME, +4.9% on LongVideoBench, and +8.8% on MLVU; even at 64 frames—where the uniform baseline is stronger—EFS maintains clear advantages. Demonstrating model-agnostic behavior (subplot b), EFS likewise benefits LLaVA-OneVision-7B and Qwen2.5-VL-7B on VideoMME, yielding +3.3% and +4.1% improvements, respectively, with 8 frames. Collectively, these results indicate that EFS is a robust, plug-and-play module that delivers substantial accuracy gains irrespective of LVLM architecture or frame budget, underscoring its practical value in diverse settings.

Table 2: Video-based question answering accuracy (%) of different Strategies in EFS.

| | Uniform | Event Partition | | Anchor Initialization | | MMR w.o. Adaptive | | | Ours |
|---|---|---|---|---|---|---|---|---|---|
| | | Random | Clustering | Random | Center | $\tau$=0.4 | $\tau$=0.6 | $\tau$=0.8 | |
| VideoMME | 60.6 | 62.9 | 63.3 | 62.1 | 61.9 | 63.7 | 63.4 | 63.6 | **64.0** |
| LongVideoBench | 57.2 | 59.8 | 60.7 | 60.3 | 60.1 | 59.9 | 61.2 | 60.9 | **61.3** |

**Ablation on EFS Strategies.** To validate the EFS pipeline, we conducted a component-wise ablation on LLaVA-Video-7B using 16 input frames (see Table 2 and Table 3).

1) Event Partitioning. In Table 2, defining event boundaries at local minima of the DINOv2 similarity curve achieves the best results, outperforming both random and clustered partitioning. This indicates that detecting natural temporal boundaries is more effective and computationally efficient than arbitrary or clustering-based divisions. Furthermore, to clarify the relationship between our method and traditional shot boundary detection, we conducted an additional experiment (Table 3). We substituted our DINOv2-based partitioning module with several established detectors from the PySceneDetect (Castellano, 2025), which operate on lower-level visual cues. Our approach again outperforms all, supporting our hypothesis: although conceptually related to shot detection, exploiting powerful self-supervised features such as DINOv2 yields a more semantically faithful and accurate partitioning of video events—capabilities that are crucial for downstream reasoning tasks.

Table 3: Video-based question answering accuracy (%) of different Shot Boundary Detectors.

|  | Uniform | Adaptive | Content | Hash | Histogram | Threshold | Ours |
|---|---|---|---|---|---|---|---|
| VideoMME | 60.0 | 63.6 | 63.6 | 62.8 | 62.7 | 63.2 | **64.0** |
| LongVideoBench | 57.2 | 60.3 | 60.4 | 60.4 | 60.3 | 59.8 | **61.3** |

2) Anchor Initialization. Next, we examine how the initial anchor frames are selected within each event. Choosing the most query-relevant frame per event clearly surpasses both random or visual centroid-based selection, confirming that semantic relevance outweighs visual representativeness for video QA. Prioritizing query relevance enables the set of anchors is strongly aligned with user intent.

3) Adaptive Refinement. Finally, we validate our adaptive refinement mechanism. As shown in Table 2, Our anchor-guided adaptive refinement consistently outperforms a standard MMR implementation with a fixed diversity threshold $\tau$, as no static threshold generalizes across both benchmarks. Dynamically adjusting the diversity criteria based on each video's unique content yields the top results (64.0/61.3%), demonstrating the robustness and effectiveness of content-adaptive refinement.

Table 4: Computational cost and accuracy comparison on the VideoMME benchmark. We report the average per-video time consumption (in seconds) and performance for the LLaVA-Video baseline (Uniform Sampling), AKS, and our proposed EFS. All experiments were conducted on a single NVIDIA A800 GPU and two Intel Xeon Silver 4314 CPUs (16 cores). All experiments were conducted on a single NVIDIA A800 GPU with two Intel Xeon Silver 4314 CPUs (16 cores). For all methods, the frame rate was fixed at 1 fps, yielding 16 input frames per video.

| Methods | Duration | Reading(s) | Signal Acquistion(s) Vision | Signal Acquistion(s) Semantics | Selection(s) | Inference(s) | Accuracy(%) |
|---|---|---|---|---|---|---|---|
| LLaVA-Video | Short | - | - | - | - | 2.39 | 72.11 |
| LLaVA-Video + AKS | Short | 5.31 | - | 13.35 | $\sim 0$ | 2.39 | 74.22(+2.11) |
| LLaVA-Video + EFS | Short | 1.45 | 1.16 | 1.34 | 0.03 | 2.39 | **76.33** (+4.22) |
| LLaVA-Video | Medium | - | - | - | - | 2.99 | 58.33 |
| LLaVA-Video + AKS | Medium | 22.61 | - | 76.28 | $\sim 0$ | 2.99 | 60.00(+1.67) |
| LLaVA-Video + EFS | Medium | 9.4 | 7.45 | 8.48 | 0.16 | 2.99 | **62.67** (+4.34) |
| LLaVA-Video | Long | - | - | - | - | 3.51 | 51.22 |
| LLaVA-Video + AKS | Long | 110.77 | - | 320.74 | $\sim 0$ | 3.51 | 53.00(+1.78) |
| LLaVA-Video + EFS | Long | 48.94 | 33.75 | 40.75 | 0.95 | 3.51 | **53.00** (+1.78) |
| LLaVA-Video | Overall | - | - | - | - | 2.97 | 60.56 |
| LLaVA-Video + AKS | Overall | 46.23 | - | 136.79 | $\sim 0$ | 2.97 | 62.41(+1.85) |
| LLaVA-Video + EFS | Overall | 19.93 | 14.12 | 16.85 | 0.38 | 2.97 | **64.00** (+3.44) |

**Efficiency and Cost-Effectiveness Analysis.** To contextualize our method's efficiency, we report its computational overhead in Table 4. Although EFS incurs a pre-processing cost absent in uniform sampling, this investment is justified by a considerable +3.44% gain in overall accuracy. More importantly, EFS is markedly more cost-effective than other advanced methods such as AKS (evaluated using its official codebase, (Tang et al., 2025)): it is over 3.5× faster in pre-processing (51.3 s vs. 183.0 s per video) while delivering nearly double the accuracy improvement (+3.44% vs. +1.85%). This balance renders EFS a practical choice for in-depth analysis where accuracy is paramount and moderate latency is acceptable, such as offline content understanding or forensic video review.

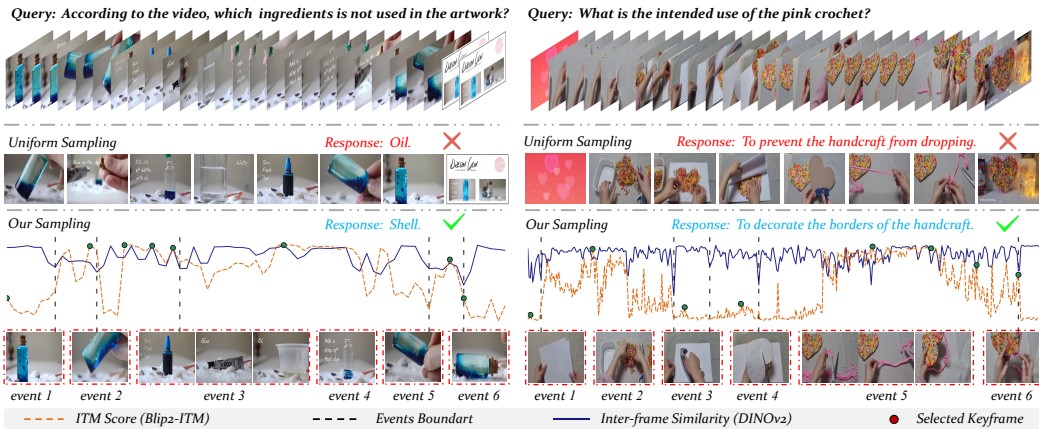

Figure 4: Qualitative comparison of our EFS versus the uniform sampling. The uniform baseline fails by missing key events, while EFS explicitly identifies the video's underlying event structure, enabling accurate reasoning and correct answers. Please zoom in for details.

Moreover, the overhead is reducible, as the main bottleneck—feature extraction—can be substantially accelerated through future optimizations, including model quantization and hardware-specific enhancements. Notably, when the input frame-per-second (fps) is moderately reduced, performance remains stable while time cost drops significantly, especially for long videos.

**Visual and semantic signal Ablations.** We further ablate the signal acquisition components of our framework (Table 5). For event boundary detection, DINOv2 consistently outperforms CLIP (Radford et al., 2021), likely due to its superior capture of fine-grained visual changes, providing a more accurate representation of visual events than the high-level semantic features of CLIP. For relevance scoring, BLIP2-ITM shows clear gains over CLIP, highlighting the benefit of dedicated image-text matching. Combining DINOv2 for event structure with

Table 5: Ablation of signal acquisition. Left of "+" denotes the visual feature extractor, while the right side is the model used for query-frame relevance.

| Methods | Frames | VideoMME | LVB(val) |
|---|---|---|---|
| *Uniform* | 8/16 | 56.2/60.6 | 54.9/57.2 |
| *CLIP + CLIP* | 8/16 | 60.2/63.1 | 57.6/60.2 |
| *DINOv2 + CLIP* | 8/16 | 60.5/63.1 | 58.4/60.3 |
| *CLIP + BLIP2-ITM* | 8/16 | 60.4/63.3 | 58.8/60.4 |
| *DINOv2 + BLIP2-ITM* | 8/16 | 60.9/64.0 | 59.8/61.3 |

BLIP2-ITM for relevance delivers the best performance, supporting our design choices.

## 4.4 VISUALIZATION

Figure 4 presents two qualitative examples from VideoMME that illustrate the advantages of our Event-Anchored Frame Selection (EFS) over uniform sampling. In both video QA cases—a bottle DIY and a crafting tutorial—the uniform baseline, being temporally agnostic, misses key events and yields incorrect answers. By contrast, EFS leverages Image–Text Matching (ITM) scores and inter-frame similarity to uncover the underlying event structure of the video content and then strategically selects frames to ensure comprehensive event coverage. As a result, the LVLM receives the necessary evidence for accurate reasoning and correct responses.

## 5 CONCLUSION

In this paper, we introduce Event-Anchored Frame Selection (EFS), a training-free frame selection method designed to boost LVLMs for long-form video understanding. By leveraging intrinsic event structure of the video, EFS constructs a keyframe representation optimized for event coverage, query relevance, and visual diversity. This event-aware approach yields significant gains across major benchmarks, confirming its superiority over flat, temporally-agnostic sampling for effective long-video reasoning. We discuss limitations and future works of EFS in Appendix B.

## ETHICS STATEMENT

Human Subjects and Data Usage. This study did not involve human subjects, nor did it require direct interaction with or intervention on individuals. Consequently, approval from an Institutional Review Board (IRB) or an equivalent ethics committee was not required. Our research was conducted exclusively on publicly available long-form video question-answering benchmarks, namely VideoMME, LongVideoBench, and MLVU, as cited in the main paper. We have not redistributed any copyrighted material or private data. All datasets and pretrained models were utilized in strict accordance with their respective licenses and terms of service.

Methodology and Data Privacy. Our proposed method, Event-Anchored Frame Selection (EFS), is a training-free, offline pre-processing module designed to identify keyframes in videos. The entire process, including feature extraction and frame selection, was conducted locally, ensuring that no video content was transmitted to external services. To protect privacy and intellectual property, we will not release any personally identifiable information (PII), raw video footage, or audio data. Our commitment to reproducibility is fulfilled through the public release of the source code and configuration files necessary to replicate our findings.

Bias and Mitigation. EFS relies on pretrained models (DINOv2 and BLIP2-ITM), which introduces the potential risk of inheriting and amplifying social or representational biases present in their training data. This could lead to misinterpretations or inequitable characterizations. To mitigate these risks, we advocate for coupling such technologies with continuous model alignment, robust content moderation mechanisms, and, in high-stakes applications, human-in-the-loop oversight. While EFS demonstrates notable advancements in video understanding, we stress that its deployment must be guided by rigorous ethical evaluation and responsible governance.

Research Integrity and Reproducibility. We provide comprehensive details on the datasets, models, hyperparameters, and evaluation protocols employed. To facilitate full replication of our results, we will release all associated code and experimental scripts. To specifically isolate and evaluate the contribution of our visual selection method, the use of subtitles was intentionally omitted from our experiments. All reported results were generated under a unified and meticulously documented experimental setup.

Conflicts of Interest. We declare no competing financial or non-financial interests that could have unduly influenced the work presented in this manuscript. Any potential conflicts that may arise will be fully disclosed in the camera-ready version.

Legal Compliance and Intellectual Property. We affirm our commitment to respecting all dataset licenses and intellectual property rights. Our methodology does not involve circumventing Digital Rights Management (DRM) or accessing content in a manner that violates access controls. Any downstream applications of this work must ensure full compliance with all relevant laws and platform-specific terms of use.

Environmental Impact. EFS introduces a moderate computational overhead during its offline pre-processing stage. We provide a detailed analysis of its timing and efficiency. By design, our preference for lightweight, training-free components aims to reduce the overall computational cost. We encourage the research community to adopt our recommended settings and utilize hardware-efficient implementations to minimize the collective energy footprint.

## REPRODUCIBILITY STATEMENT

For reproducibility, we provide detailed descriptions of our methods and experimental setups in Section 4.1 and Appendix C. We further validate the robustness of our findings with additional experiments (Figure 3). To facilitate replication, the code is included in the supplementary material and will be released publicly.

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

## A    STATEMENT ON LARGE LANGUAGE MODEL (LLM) USAGE

In accordance with the ICLR 2026 policy on the use of Large Language Models (LLMs), we disclose that an LLM was utilized as a general-purpose writing assistance tool in the preparation of this manuscript. The use of the LLM was strictly limited to post-writing text refinement. Specifically, its role included:

- Grammar and Spelling Correction: Identifying and correcting grammatical errors and typos.
- Improving Clarity and Diction: Suggesting alternative phrasing, refining word choices, and improving the overall readability of sentences and paragraphs that were already written by the authors.
- LaTeX Formatting: Assisting with minor LaTeX code adjustments and debugging for formatting purposes.

Crucially, the LLM was not used for research ideation, generating scientific claims, producing experimental results, or drafting any substantive parts of the paper from scratch. All concepts, methodologies, analyses, and conclusions presented in this work are entirely those of the human authors. The authors have carefully reviewed and edited all LLM-suggested modifications and take full responsibility for the scientific integrity and accuracy of the final submitted content.

## B    LIMITATIONS AND FUTURE WORK

### B.1    LIMITATIONS

Despite its demonstrated efficacy, our Event-Anchored Frame Selection (EFS) framework is subject to several inherent limitations that warrant discussion.

- **Preprocessing Overhead.** Preprocessing Overhead. Our pipeline requires an offline signal extraction stage based on pretrained DINOv2 (Oquab et al., 2023) and BLIP2-ITM (Li et al., 2023), incurring computational costs that uniform sampling avoids. For long videos, generating these signals constitutes a substantial preprocessing burden, which can render EFS impractical in latency-sensitive or real-time settings.
- **Heuristic and Hyperparameter Sensitivity.** EFS is a multi-stage, heuristic-driven pipeline whose performance hinges on several key hyperparameters, most notably the number of events $M$ and the relaxation factor ($\alpha$). Although we identify workable configurations via grid search, these settings may not transfer across video domains or downstream tasks. The current framework lacks a principled, data-driven procedure for automatic hyperparameter adaptation, leaving performance sensitive to manual choices.
- **Dependence on Upstream Models.** The performance of EFS is intrinsically coupled with the representational power of the foundational DINOv2 and BLIP2-ITM models. Consequently, EFS inherits not only their capabilities but also their limitations. Any inherent biases, domain gaps, or catastrophic failures within these upstream models can propagate downstream, adversely affecting the quality of the final frame selection.

### B.2    FUTURE WORKS

The limitations of our current approach highlight several promising directions for future research.

- **System-Level Efficiency Enhancement.** A primary objective is to reduce the end-to-end latency of the proposed framework. Future work will focus on holistic system-level optimizations across the entire pipeline. This includes accelerating data ingestion and processing, designing more efficient computational pathways, and exploring hardware-aware execution strategies to ensure stable, low-latency performance under strict resource constraints without compromising selection quality.
- **Omni-Modal Event Understanding.** To achieve a more robust and context-aware analysis, we plan to move beyond purely visual cues and develop an omni-modal framework. This involves integrating diverse data streams—such as high-level semantics, motion dynamics, audio signals,

and textual transcriptions—into a unified representation. By fusing these multimodal signals, the model can learn to identify complex event boundaries and resolve ambiguities that are imperceptible from visual information alone, leading to a more holistic understanding of the video content.

- **End-to-End Trainable Selection.** A key ambition is to replace the current heuristic-based pipeline with a fully end-to-end trainable model. This would involve designing a differentiable selection module that learns to identify key content by directly optimizing for the downstream task objective. Such a data-driven approach would not only eliminate the need for manual hyperparameter tuning but also has the potential to discover more effective and adaptive selection strategies than a handcrafted pipeline, ultimately yielding a more powerful and generalizable solution.

## C MORE DETAILS OF EXPERIMENTAL SETUPS

### C.1 MODELS AND CODEBASE

Our experiments leverage several public models and toolkits. Their official resources or code are listed below.

- **BLIP2-ITM-ViT-g** (Li et al., 2023): https://huggingface.co/Salesforce/blip2-itm-vit-g
- **DINOv2-base** (Oquab et al., 2023): https://huggingface.co/facebook/dinov2-base
- **LLaVA-OneVision-7B** (Li et al., 2024a): https://huggingface.co/lmms-lab/llava-onevision-qwen2-7b-ov
- **LLaVA-Video-7B** (Zhang et al., 2024c): https://huggingface.co/lmms-lab/LLaVA-Video-7B-Qwen2
- **Qwen2.5-VL-7B** (Bai et al., 2025): https://huggingface.co/Qwen/Qwen2.5-VL-7B-Instruct
- **LMMs-Eval toolkit** (Zhang et al., 2024a): https://github.com/EvolvingLMMs-Lab/lmms-eval

### C.2 HYPERPARAMETERS SETTINGS

The optimal hyperparameters for the number of events (M) and the threshold relaxation factor ($\alpha$) vary depending on the dataset and the number of input frames. Table 6 summarizes the settings used in our experiments. A detailed ablation study on these hyperparameters is provided in D.3.

Table 6: Hyperparameter settings for different datasets and frame counts.

| Frames | VideoMME | LongVideoBench | MLVU |
|---|---|---|---|
| 8 | M=6, $\alpha$=0.7 | M=6, $\alpha$=0.3 | M=6, $\alpha$=0.4 |
| 16 | M=10, $\alpha$=0.5 | M=14, $\alpha$=0.1 | M=12, $\alpha$=0.1 |
| 32 | M=28, $\alpha$=0.3 | M=20, $\alpha$=0.4 | M=16, $\alpha$=0.4 |
| 64 | M=32, $\alpha$=0.7 | M=44, $\alpha$=0.3 | M=44, $\alpha$=0.4 |

## D MORE EXPERIMENTAL RESULTS

### D.1 COMPARISONS WITH DIFFERENT SUBTASKS IN VIDEOMME

For a fine-grained evaluation of our Event-Anchored Frame Selection (EFS) method, we conducted a comprehensive comparative analysis against a uniform sampling baseline on the VideoMME benchmark. Both methods were evaluated using the LLaVA-Video-7B model, with the input constrained to eight frames. This setup allowed for a detailed assessment of performance across all 12 subtasks, revealing the specific domains where EFS provides the most significant advantages.

As illustrated in Figure 5, EFS demonstrates superior performance to the baseline across every subtask, underscoring its broad-spectrum efficacy rather than a narrow, task-specific benefit. The most substantial gains are observed in tasks demanding high perceptual acuity and temporal awareness, including Spatial Perception (+11.11%), Temporal Perception (+9.09%), and OCR Problems

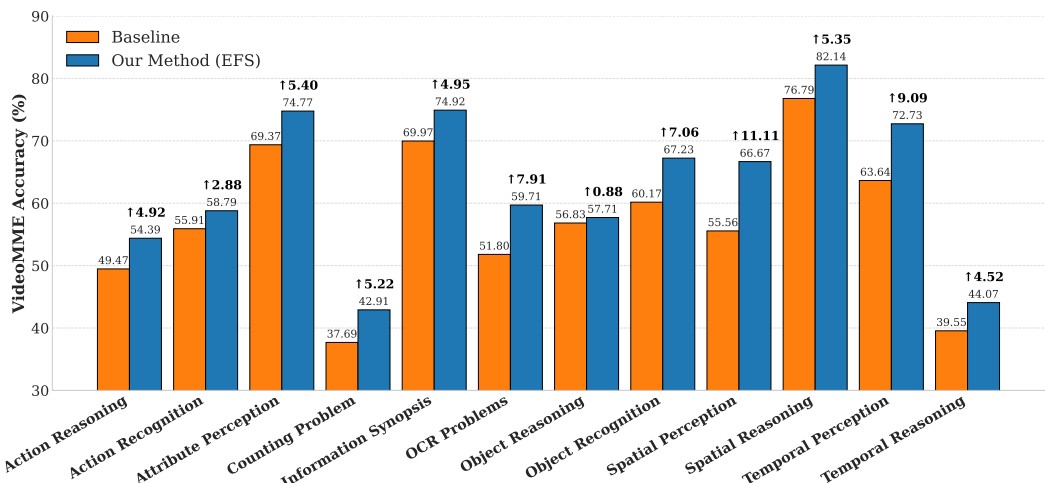

Figure 5: Detailed performance comparison of our method (EFS) against the baseline across all 12 subtasks of the Video-MME benchmark (Fu et al., 2025). The results demonstrate that our EFS approach consistently surpasses the baseline in every category, underscoring the broad effectiveness of our method in enhancing diverse video understanding capabilities.

(+7.91%). This highlights EFS's strength in capturing and preserving essential scene context, sequential order, and fine-grained visual details. Moreover, notable improvements in complex, high-level tasks such as Spatial Reasoning (+5.35%), Action Reasoning (+4.92%), and Temporal Reasoning (+4.52%) further affirm the generalizability of our approach. The consistent performance uplift across this diverse range of tasks, from low-level perception to advanced reasoning, validates EFS as a robust and effective framework for long-video understanding, particularly under the practical limitation of a finite context window.

Table 7: Ablation on the window size $l$. We report the video-based question answering accuracy (%) on the VideoMME, LVB, and MLVU benchmarks. The best results are highlighted in **bold**.

| Method | VideoMME | LVB(val) | MLVU |
|---|---|---|---|
| *uniform* | 60.6 | 57.2 | 60.9 |
| *EFS(l=3)* | **64.0** | 61.3 | **69.0** |
| *EFS(l=5)* | 63.3 | **61.7** | 68.0 |
| *EFS(l=7)* | 63.4 | 60.9 | 68.6 |

## D.2 ABLATION STUDY ON WINDOW SIZE

The window size, denoted as $l$, is a critical hyperparameter in our Event-Anchored Frame Selection (EFS) framework, as it defines the temporal scope for computing inter-frame similarity. The choice of $l$ involves a direct trade-off: a smaller window enhances sensitivity to rapid, fine-grained scene changes, while a larger one provides a more robust measure of local temporal coherence. To systematically investigate this trade-off and determine an optimal configuration, we conducted an ablation study with $l$ set to 3, 5, and 7. The performance of each variant was evaluated against a uniform sampling baseline across three diverse video understanding benchmarks: VideoMME (Fu et al., 2025), LongVideoBench (LVB)(Wu et al., 2024), and MLVU(Zhou et al., 2024).

As summarized in Table 7, all configurations of EFS substantially outperform the uniform sampling baseline, affirming the fundamental advantage of content-aware selection. Among the tested variants, a window size of $l$=3 yields the strongest overall performance, achieving state-of-the-art results of 64.0% on VideoMME and 69.0% on MLVU. Although the $l$=5 setting shows a marginal advantage on the LVB validation set, its performance on the other benchmarks declines. Increasing the window to $l$=7 leads to a more pronounced degradation in accuracy. These results strongly indicate

that a compact window ($l$=3) is most effective at capturing the salient, short-term visual dynamics crucial for downstream comprehension tasks. In contrast, larger windows risk over-smoothing the temporal signal, thereby obscuring important transitional moments. Consequently, we adopt $l$=3 as the default window size for our EFS framework in all subsequent experiments.

### D.3 HYPERPARAMETER ABLATION STUDY

To dissect the contributions of our framework's key components, we conduct a comprehensive ablation study on two critical hyperparameters: the maximum number of video events, $M$, and the threshold relaxation factor, $\alpha$, used in our adaptive MMR algorithm. We perform a grid search on these parameters and report the accuracy on the VideoMME, LongVideoBench (LVB), and MLVU benchmarks.

**Impact of Event Granularity** ($M$). The hyperparameter $M$ dictates the granularity of the initial video partition. An optimal value for $M$ must strike a balance: if $M$ is too small, distinct semantic events may be merged, leading to a coarse-grained perception and a poor set of initial anchors. Conversely, if $M$ is excessively large, a single event might be over-partitioned, introducing noisy or redundant anchors that undermine the subsequent refinement process.

As shown in Table 8, our empirical results on VideoMME corroborate this trade-off. Performance steadily improves as $M$ increases from 1 to 10, peaking at an accuracy of **64.0%**. However, further increasing $M$ to 12 and beyond leads to a marginal but consistent decline in performance. A similar trend is observed on the LongVideoBench and MLVU datasets (Table 9), where the performance peaks when $M$ is in the range of 12 to 14. This consistent pattern across benchmarks validates our hypothesis that an intermediate event granularity is crucial for building a representative and effective structural prior for frame selection.

Table 8: Ablation of $M$ and $\alpha$ on the VideoMME benchmark.

| $M$ \ $\alpha$ | 0.1 | 0.3 | 0.4 | 0.5 | 0.6 | 0.7 | 0.9 |
|---|---|---|---|---|---|---|---|
| 1 | 61.4 | 61.3 | 61.0 | 61.5 | 61.7 | 61.9 | 61.6 |
| 4 | 62.5 | 62.4 | 62.4 | 62.5 | 62.2 | 61.9 | 61.4 |
| 8 | 62.6 | 62.9 | 62.8 | 62.7 | 62.4 | 62.6 | 62.2 |
| 10 | 63.0 | 63.4 | 63.8 | **64.0** | 63.6 | 63.4 | 63.5 |
| 12 | 63.0 | 63.1 | 63.1 | 63.3 | 62.9 | 63.4 | 63.4 |
| 14 | 63.4 | 63.2 | 63.3 | 63.4 | 62.9 | 63.2 | 62.7 |
| 16 | 63.1 | 63.0 | 63.1 | 63.2 | 63.0 | 63.2 | 63.0 |

**Impact of Threshold Relaxation Factor** ($\alpha$). The relaxation factor, $\alpha$, controls the flexibility of the diversity thresholds in our adaptive MMR framework. A small $\alpha$ enforces a narrow and rigid diversity range, which may fail to adapt to the video's intrinsic visual variance. In contrast, a large $\alpha$ makes the similarity threshold overly loose (or strict, depending on formulation), potentially filtering out relevant frames and thus reducing coverage.

Interestingly, the optimal value for $\alpha$ exhibits a dataset-specific nature. For VideoMME (Table 8), a moderate $\alpha$=0.5 achieves the best balance, enabling the model to dynamically adjust to the video's statistical characteristics. However, for both LongVideoBench and MLVU (Table 9), a much smaller value of $\alpha$=0.1 consistently yields the best results. This suggests that the videos in LVB and MLVU may possess more uniform event structures or statistical properties, benefiting from a stricter and less adaptive diversity criterion. The optimal configuration for LVB is ($M$=14, $\alpha$=0.1), achieving **61.3%**, and for MLVU is ($M$=14, $\alpha$=0.1), achieving **69.0%**.

These findings underscore that while the general trends of our hyperparameters are consistent, their optimal values are influenced by the unique statistics and event structures of each dataset. This highlights the importance of data-aware priors in guiding the frame selection process effectively.

Table 9: Ablation of $M$ and $\alpha$. Results are shown as accuracy (%) on LongVideoBench/MLVU.

| $M$ \ $\alpha$ | 0.1 | 0.3 | 0.5 | 0.7 |
|---|---|---|---|---|
| 1 | 59.5/68.2 | 58.7/68.0 | 58.3/67.6 | 58.8/67.6 |
| 4 | 60.0/67.9 | 59.6/67.6 | 59.8/68.3 | 59.6/67.8 |
| 8 | 59.7/68.0 | 60.0/68.0 | 59.7/67.5 | 59.0/67.9 |
| 10 | 60.0/68.6 | 60.0/68.5 | 60.3/67.7 | 60.4/67.7 |
| 12 | 60.6/**69.0** | 59.8/68.0 | 60.4/68.0 | 59.9/68.1 |
| 14 | **61.3/69.0** | 60.4/68.6 | 61.1/68.5 | 60.3/68.7 |
| 16 | 60.1/67.8 | 59.9/67.8 | 59.9/67.8 | 60.1/67.8 |

### D.4 GENERALIZABILITY TO INFORMATION SYNOPSIS QUERIES

Our EFS method is fundamentally query-based, which raises a question about its effectiveness on information synopsis queries (e.g., "What does this video describe?"). Intuitively, for such general questions where specific semantic signals are absent, the query-centric anchor selection of EFS might not be beneficial, and a simpler uniform sampling approach could suffice.

To investigate this, we conducted an experiment to determine if an LVLM could dynamically choose the appropriate sampling strategy. We implemented a Chain-of-Thought (CoT) based router: for each query, the LLaVA-Video model was first prompted to classify the query as either specific and targeted, or general and synopsis-oriented. The exact prompt used for this classification is detailed in D.5. If the model identified a query as specific, we applied our EFS method. If it deemed the query general, the system reverted to a standard uniform sampling strategy. We refer to this hybrid approach as "part EFS". All experiments were conducted with a 16-frame input limit.

The results are presented in Table 10. Counter-intuitively, the adaptive "part EFS" strategy consistently underperformed the full EFS approach across all three benchmarks. For instance, on VideoMME, reverting to uniform sampling for general queries resulted in a 1.5% drop in accuracy compared to always using EFS. This performance degradation demonstrates that forcing a fallback to uniform sampling for general queries is a detrimental choice. The result strongly suggests that the structural priors and enhanced event coverage provided by EFS are beneficial even for information synopsis tasks, outperforming the temporally-agnostic nature of uniform sampling. This conclusion is further supported by the performance gains observed on the Information Synopsis sub-task, as illustrated in Figure 5.

Table 10: Performance comparison of the CoT-based adaptive sampling strategy. The performance drop in "part EFS" highlights the universal effectiveness of EFS, even on general queries where the system reverted to uniform sampling.

| Methods | VideoMME | LVB | MLVU |
|---|---|---|---|
| LLaVA-Video (Uniform) | 60.6 | 57.2 | 60.9 |
| LLaVA-Video + EFS | 64.0 | 61.3 | 69.0 |
| LLaVA-Video + part EFS | 62.5(-1.5) | 60.7(-0.6) | 68.6(-0.4) |

### D.5 ADDITIONAL QUALITATIVE EXAMPLES

To further validate the rationality of our event partitioning, Figure 6 presents a t-SNE visualization of DINOv2 features extracted from sampled video frames. The clear clustering and separation of points confirm that our method groups visually consistent events and identifies meaningful event boundaries. This establishes a reliable foundation for subsequent keyframe selection.

972
973
974
975
976
977
978
979
980
981
982
983
984
985
986
987
988
989
990
991
992
993
994
995
996
997
998
999
1000
1001
1002
1003

**System Prompt: Video Query Classifier**

You are an expert assistant in video analysis. Your task is to determine if a user's question necessitates a query-based keyframe selection method. This method is designed for finding specific moments, objects, or actions within a video.

Analyze the user's question. Think step-by-step to decide if it's a specific query or a general one. After your reasoning, provide your final answer on a new line. The final answer must be ONLY 'Yes' or 'No'.

'Yes' means the question requires a specific, targeted search. 'No' means the question is general, asking for a summary or an overall idea.

**Example 1:**
Question: "Summarize what this video is about."
Reasoning: This is a high-level request for a general overview. It does not require locating a specific moment.
Final Answer: No

**Example 2:**
Question: "Find the scene where the cat knocks over the vase."
Reasoning: This asks to locate a very specific action ("knocks over the vase") involving specific subjects ("cat", "vase"). This requires a targeted search.
Final Answer: Yes

**Example 3:**
Question: "What is the overall sentiment of the speaker?"
Reasoning: This question is about the general tone throughout the video, not a single instant. It is a holistic analysis.
Final Answer: No

**Example 4:**
Question: "When does the CEO appear on screen?"
Reasoning: This requires identifying all specific moments when a particular person ("the CEO") is visible. This is a query-specific task.
Final Answer: Yes

**Your Task:**
Question: {Your Question}
Reasoning:

1004
1005
1006
1007
1008
1009
1010
1011
1012
1013
1014
1015
1016
1017

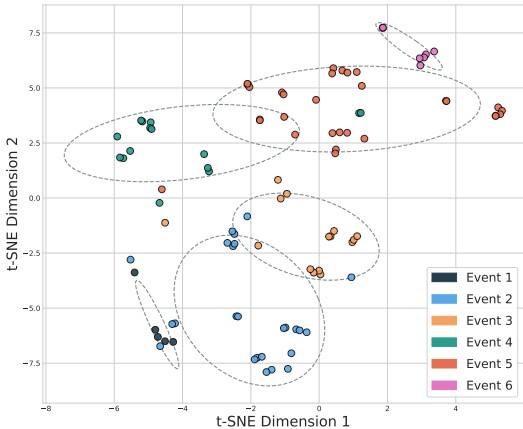

Figure 6: T-SNE visualization of DINOv2 features for sampled video frames. Each point represents one frame, and color indicates the event predicted by our EFS pipeline.
