# OpenReview forum: "Event-Anchored Frame Selection for Efficient Long-Video Understanding"
_ICLR.cc/2026/Conference — ICLR 2026 Conference Withdrawn Submission_

### Official Review · Reviewer_RzF8 · 2025-10-24

**Soundness:** 3
**Presentation:** 3
**Contribution:** 2
**Rating:** 4
**Confidence:** 4

**Summary:**

This paper presents Event-Anchored Frame Selection (EFS), a training-free and plug-and-play pipeline for keyframe selection in long-video understanding. The method first segments a video into visually homogeneous events via DINOv2 embeddings, then selects query-relevant “anchor” frames using BLIP2-ITM, and finally performs an adaptive Maximal Marginal Relevance (MMR) refinement to balance coverage, relevance, and diversity. EFS is evaluated across three long-video QA benchmarks (VideoMME, LongVideoBench, MLVU) and demonstrates consistent accuracy gains for multiple LVLMs (LLaVA-Video, LLaVA-OneVision, Qwen2.5-VL). The approach is efficient, modular, and empirically robust.

**Strengths:**

- **Well-structured pipeline.** The paper combines unsupervised visual segmentation, query-aware anchor selection, and adaptive refinement in a coherent multi-stage framework.

- **Training-free and efficient.** The method requires no fine-tuning, making it easily deployable to various LVLMs.

- **Extensive empirical validation.** Results cover multiple benchmarks, models, and frame budgets, with detailed ablations and qualitative visualizations.

- **Clear presentation.** The paper is clearly written with strong figures (e.g., Fig. 1–4) and thorough experimental setups.

**Weaknesses:**

- **Limited Novelty and Incremental Contribution.** While EFS is elegantly engineered, its components—event segmentation, anchor-based selection, and MMR refinement—are individually well-established. Similar “event-aware” or structure-guided frame selection frameworks have appeared in AKS (Tang et al., 2025), $T^{*}$ (Ye et al., 2025), Logic‑in‑Frames (Guo et al., 2025), mDP3 (Sun et al., 2025), VSI (He et al., 2025) and Nar-KFC (Fang et al., 2025). The adaptive MMR is a useful tweak but not a conceptual breakthrough. The overall framework feels like a careful integration rather than a fundamentally new idea.

- **Lack of Theoretical or Analytical Insight.** The method is largely empirical and heuristic. There is no principled analysis or theoretical grounding to justify why event segmentation + adaptive MMR leads to better reasoning beyond intuitive explanations. A formal examination of coverage–diversity trade-offs or selection optimality is missing.

- **Heavy Reliance on Handcrafted Hyperparameters.** The approach depends on manually chosen values for event number 𝑀, relaxation factor 𝛼, and fixed frame sampling rate (1 fps). These hyperparameters are dataset-specific (Appendix C.2), raising concerns about reproducibility and generalization to other video domains or tasks.

- **No End-to-End Learning or Adaptivity.** Although the training-free nature is a strength for efficiency, it also restricts adaptivity. The pipeline cannot learn task- or model-specific cues, limiting scalability to more complex or diverse LVLM architectures. The authors themselves acknowledge this in the “Future Work” section (B.2).

- **Ambiguous Distinction from Shot Detection.** The “event partitioning” step relies on detecting local minima of temporal similarity curves, which is conceptually close to traditional shot boundary detection. While Table 3 provides some empirical differentiation, the conceptual novelty over shot detection remains insufficiently justified.

- **Limited Exploration Beyond QA.** The paper’s experiments focus exclusively on multiple-choice video QA. The method’s claimed generality (e.g., for captioning, summarization, or temporal reasoning) is not demonstrated, leaving the broader utility speculative.

- **Missing Analysis on Failure Cases and Selection Behavior.** While qualitative examples are shown, there is no systematic analysis of when EFS fails—e.g., for videos with smooth transitions, repetitive scenes, or motion-heavy segments where visual similarity is unreliable. Understanding such behaviors would strengthen the paper’s empirical claims.

- **Inter-module Independence and Information Loss.** The three stages (DINOv2 partition → BLIP2-ITM anchor → adaptive MMR) are executed sequentially with no feedback loop. Errors from earlier stages (e.g., missegmented events or irrelevant anchors) can propagate without correction. An end-to-end or iterative variant might better capture dependencies.

- **No Statistical Significance or Variance Reporting.** The paper reports average accuracies but lacks variance, confidence intervals, or statistical tests across runs. This makes it hard to judge the reliability of reported 1–2% gains, especially for large models with stochastic inference.

**Questions:**

1. How does EFS perform when applied to continuous motion videos (e.g., sports, surveillance) where event boundaries are less distinct?
2. Could the adaptive threshold in MMR be learned or calibrated dynamically instead of being fixed per dataset?
3. Is there any performance degradation when substituting DINOv2 or BLIP2-ITM with other encoders (e.g., CLIP-L/14)?
4. How sensitive is the performance to the choice of 𝑀 and 𝛼? Can these be auto-tuned?
5. Have the authors evaluated whether event segmentation correlates with semantic event changes rather than just visual discontinuities?

---

### Official Review · Reviewer_GB4z · 2025-10-29

**Soundness:** 3
**Presentation:** 3
**Contribution:** 2
**Rating:** 6
**Confidence:** 3

**Summary:**

This paper proposes an event-anchored frame selection method, termed as EFS. EFS first detects events using DINO features of consecutive frames. Within each detected event, key frames are subsequently selected using a VLM. To enhance diversity, an adaptive Maximal Marginal Relevance (MMR) mechanism is then applied. The whole pipeline is training free and plug and play, show clear improvements on three commonly used long video QA benchmarks. Moreover, EFS provides notable efficiency and performance gains compared to prior frame selection methods.

**Strengths:**

1. The idea of anchoring frame selection on detected events is intuitive. Addressing the challenge of selecting representative frames in very long videos, event detection serves as a meaningful precursor for identifying key segments.
2. The efficiency of the proposed approach is particularly impressive. As shown in Table 4, EFS significantly reduces preprocessing time relative to previous methods, while simultaneously achieving superior performance.

**Weaknesses:**

1. While EFS shares insights with prior methods such as **BOLT**, **AKS**, and **NFC**, which also target relevance and diversity in selected frames, the distinctive advantages of EFS over these approaches remain insufficiently discussed.
2. Although Table 4 highlights clear efficiency gains relative to AKS, the underlying factors contributing to this improvement are not adequately analyzed.
3. The paper omits discussion of Q-Frame, a recent baseline that also leverages query–frame relevance for selection.
4. The core contribution—event-anchored frame selection—is primarily evaluated on general video QA benchmarks. It remains uncertain whether EFS preserves key events effectively in event-centric tasks. Incorporating evaluation on event-focused datasets (e.g., grounded video QA [1], video grounding, or dense video captioning) would better validate the proposed framework’s strengths.

    [1], Di et al. Grounded Question-Answering in Long Egocentric Videos. CVPR25

**Questions:**

1. The main concern lies in understanding why EFS yields more diverse and query-relevant frames compared to prior methods. What are the key differences that enable this improvement?
2. Would EFS better than other frame selection methods on event centric tasks ?
3. The method assumes single-query relevance. In real-world applications, LVLMs are often used in multi-turn dialogues, where subsequent questions differ from the initial one. Has the method been evaluated under multi-turn or conversation-driven settings to verify its robustness across query variations?

---

### Official Review · Reviewer_jeUm · 2025-11-01

**Soundness:** 2
**Presentation:** 3
**Contribution:** 3
**Rating:** 4
**Confidence:** 4

**Summary:**

This paper proposes Event-Anchored Frame Selection (EFS), a training-free hierarchical framework for keyframe selection in long-video understanding. The core contribution is a three-stage pipeline that: (1) partitions videos into visually coherent events using DINOv2 embeddings, (2) selects the most query-relevant frame from each event as an "anchor", and (3) performs anchor-guided global refinement via adaptive Maximal Marginal Relevance to balance event coverage, query relevance, and visual diversity. As a plug-and-play module, EFS significantly boosts various LVLMs' performance on long-video QA benchmarks (e.g., +4.7% on VideoMME, +4.9% on LongVideoBench, +8.8% on MLVU for LLaVA-Video-7B), demonstrating that event-aware selection substantially outperforms flat sampling paradigms.

**Strengths:**

1. Paradigm Shift to Event-Aware Frame Selection
Transitions from traditional flat sampling to a hierarchical event-aware framework, emphasizing the narrative structure and event boundaries of videos—a conceptually novel direction.

2. Training-Free and Plug-and-Play Design: EFS serves as a standalone preprocessing module that integrates seamlessly with existing LVLMs without fine-tuning
3. Adaptive Diversity Control Mechanism: Introduces an anchor-guided adaptive MMR strategy that dynamically adjusts diversity thresholds based on video content.

4. Demonstrates consistent and notable improvements across multiple LVLMs on three challenging long-video QA benchmarks—VideoMME, LongVideoBench, and MLVU—confirming broad applicability and effectiveness.

**Weaknesses:**

Indirect frame selection evaluation - Only uses downstream QA accuracy, lacking direct metrics (temporal IoU, coverage/redundancy scores) to validate whether selected frames truly correspond to key moments.

Limited comparison scope - Missing comparisons with recent segment-based methods (e.g., K-frames), making it difficult to isolate whether advantages come from event partitioning or redundancy reduction.

Insufficient connection to temporal grounding - The anchor selection mechanism shares similarities with temporal moment retrieval methods, but these connections remain unexplored in related work.

**Questions:**

See Weaknesses

---

### Official Review · Reviewer_cumw · 2025-11-02

**Soundness:** 3
**Presentation:** 2
**Contribution:** 3
**Rating:** 4
**Confidence:** 4

**Summary:**

This paper proposes Event-Anchored Frame Selection (EFS), a training-free method for selecting keyframes from long videos to improve the performance of LVLMs. The key idea is to move beyond "flat" sampling strategies by first understanding the video's event structure. EFS works in three steps: it (1) partitions the video into visually coherent events, (2) selects the most query-relevant frame from each event as an "anchor," and (3) performs a global refinement to add diverse frames. Extensive experiments on three long-video QA benchmarks show that EFS consistently and significantly boosts the accuracy of various LVLMs, especially when the frame budget is small.

**Strengths:**

- The paper tackles a practical bottleneck in long-video understanding, that is, efficient and informative frame selection. The focus on leveraging event structure is a logical approach.
- EFS is training-free and plug-and-play, making it practical.
- The experimental results convincingly demonstrate its effectiveness across multiple models and datasets.

**Weaknesses:**

- Novelty. The novelty of the proposed event partitioning method seems somewhat overstated. Since similarity scores from Dinov2 may not fully capture semantic-level changes (i.e., shifts in action or plot), the method appears to focus mainly on low-level changes like camera cuts and obvious scene transitions. As Figure 4 suggests, the partitioning is primarily based on simple, low-level units rather than high-level events. I would suggest that the authors visualize more examples to clarify the partitioning process. Moreover, the main contribution to the final performance seems to stem from BLIP2-ITM rather than the event partitioning. This is supported by Table 2, which shows that a random partition still achieves good performance, while the random anchor baseline performs poorly.
- Computational cost. The preprocessing step (feature extraction with DINOv2 and BLIP2) introduces more latency, which may be a concern for real-time applications despite being an offline cost. For hours-long videos, the problem becomes crucial.
- Moreover, the performance is tied to potential biases of DINOv2 and BLIP2.
- Hyperparameter sensitivity. The method relies on several hyperparameters (e.g., event number M, relaxation factor α) whose optimal values vary across datasets. This requires tuning and could hinder out-of-the-box usability. How to adaptively determine such factors?
- Limited gains for genral query. While the method excels on specific queries, its advantages for very general query is limited like "summarize this video", as noted in the appendix.

**Questions:**

see weaknesses

---

### Note · Authors · 2025-11-13

I have read and agree with the venue's withdrawal policy on behalf of myself and my co-authors.